

# Micromechanical modeling of snow failure

Grégoire Bobillier[1], Bastian Bergfeld[1], Achille Capelli[1], Jürg Dual[2], Johan Gaume[1,3], Alec van Herwijnen[1], Jürg Schweizer[1]

[1] WSL Institute for Snow and Avalanche Research SLF, Davos, Switzerland
[2] Institute for Mechanical Systems, ETH Zurich, Switzerland
[3] SLAB Snow and Avalanche Simulation Laboratory, EPFL Swiss Federal Institute of Technology, Lausanne, Switzerland

*Correspondence to: Grégoire Bobillier (gregoire.bobillier@slf.ch)*

**Abstract.** Dry-snow slab avalanches start with the formation of a local failure in a highly porous weak layer underlying a cohesive snow slab. If followed by rapid crack propagation within the weak layer and finally a tensile fracture through the slab appears, a slab avalanche releases. While the basic concepts of avalanche release are relatively well understood, performing fracture experiments in the lab or in the field can be difficult due to the fragile nature of weak snow layers. Numerical simulations are a valuable tool for the study of micromechanical processes that lead to failure in snow. We used a three dimensional discrete element method (3D-DEM) to simulate and analyze failure processes in snow. Cohesive and cohesionless ballistic deposition allowed us to reproduce porous weak layers and dense cohesive snow slabs, respectively. To analyze the micromechanical behavior at the scale of the snowpack (~1 m), the particle size was chosen as a compromise between a low computational cost and a detailed representation of important micromechanical processes. The 3D-DEM snow model allowed reproducing the macroscopic behavior observed during compression and mixed-modes loading of dry snow slab and weak snow layer. To be able to reproduce the range of snow behavior (elastic modulus, strength), relations between DEM particle/contact parameters and macroscopic behavior were established. Numerical load-controlled failure experiments were performed on small samples and compared to results from load-controlled laboratory tests. Overall, our results show that the discrete element method allows to realistically simulate snow failure processes. Furthermore, the presented snow model seems appropriate for comprehensively studying how the mechanical properties of slab and weak layer influence crack propagation preceding avalanche release.

## 1 Introduction

Dry-snow slab avalanches require initiation and propagation of a crack in a weak snow layer buried below cohesive slab layers. Crack propagation occurs if the initial zone of damage in the weak layer is larger than the so-called critical crack size. Weak layer fracture during crack propagation is generally accompanied by the structural collapse of the weak layer due to the high porosity of snow (van Herwijnen et al., 2010). If the crack propagates across a steep slope, a slab avalanche may release (McClung, 1979;Schweizer et al., 2003). Our understanding of crack propagation was greatly improved by the introduction of



the Propagation Saw Test (PST; Gauthier and Jamieson, 2006;Sigrist and Schweizer, 2007;van Herwijnen and Jamieson, 2005). The PST allows analyzing crack propagation propensity and deriving mechanical properties, for example, by particle tracking velocimetry (van Herwijnen et al., 2016).

The essential mechanical properties related to the onset of crack propagation are slab elasticity, slab load and tensile strength,
as well as the weak layer strength and specific fracture energy (e.g., Reuter and Schweizer, 2018). However, no theoretical framework exists that describes how these mechanical properties – and possibly other ones – relate to the dynamics of crack propagation at the slope scale. Whereas field experiments are difficult to perform at this scale, numerical simulations may provide insight into the drivers of propagation dynamics.

Johnson and Hopkins (2005) were the first to apply the discrete element method (DEM) to model snow deformation. They
simulated creep settlement of snow samples, which consisted of a 1000 randomly oriented cylinders of random length with hemispherical ends. More recently, DEM was used to model the mechanical behavior based on the complete 3-D microstructure of snow (Hagenmuller et al., 2015). Gaume et al. (2015) developed a discrete element model to simulate crack propagation and subsequently derived a new analytical expression for the critical crack length (Gaume et al., 2017b). Their approach allows generating highly porous samples and was used to perform 2-D simulations of the PST in agreement with
field experiments. However, the oversimplified shape (triangular structure) and the 2-D character of the weak layer employed by Gaume et al. (2015) prevented a detailed analysis of the internal stresses during crack propagation. On the other hand, microstructure-based DEM models adequately reproduce the mechanical behavior (Mede et al., 2018). However, the computational costs of these complex 3D-models are too high to generate samples large enough to investigate the dynamics of crack propagation at the slope scale.

Our aim is to develop a 3-D DEM snow model that adequately takes into account snow microstructure, but is not too costly in terms of computational power so that simulations at the slope scale become feasible. To relate DEM parameters to macroscopic snow behavior we will validate the model by simulating basic load cases. Finally, we numerically simulate mixed-mode loading experiments and compare results to those obtained during laboratory experiments.

## 2   Data and methods

### 2.1   Formulation of the model

*Discrete element method*

To simulate the failure behavior of layered snow samples, we used the discrete element method (DEM). DEM, first introduced by Cundall and Strack (1979) is a numerical tool, commonly employed to study granular-like assemblies composed of a large number of discrete interacting particles. We used the PFC3D (v5) software developed by Itasca Consulting Group
(http://www.itascacg.com).



*Contact model*

We used the parallel-bond contact model (PBM) introduced by Potyondy and Cundall (2004). PBM provides the mechanical behavior of a finite-sized piece of cement-like material that connects two particles. The PBM component acts in parallel with a classical linear contact model and establishes an elastic interaction between the particles. The mechanical parameters include

the contact elastic modulus $E_u$, Poisson's ratio $v_u = 0.3$ , the restitution coefficient $e_u = 0.1$, and the friction coefficient $\mu_u = 0.2$. If particles are bonded, the bond part will act in parallel to the contact part. The bonded part is described by the bond elastic modulus $E_b$, the bond Poisson's ratio $v_b = 0.3$ and the bond strength, shear and tensile strength $\sigma_s$ and $\sigma_t$. To reduce the number of variables we assume $E_u = E_b \triangleq E_{particle}$ and $\sigma_s = \sigma_t \triangleq \sigma_{bond}^{th}$. More details about the PBM can be found in previous studies (Gaume et al., 2017a;Gaume et al., 2015;Gaume et al., 2017b).

*System generation*

The simulated three dimensional system consisted of a rigid basal layer (Figure 1, blue particles), the layer studied (weak layer or slab layer, green particles), and an 'actuator' layer used to apply the load (red particles). The basal layer is composed of a single layer of particles with a radius of $r = 5$ mm. The weak layer was created by cohesive ballistic deposition (Löwe et al., 2007) to reproduce the porous and highly anisotropic structure of natural weak layers. Doing so, we obtained a porosity of

80% for a particle radius of $r = 2.5$ mm. The layer thickness (3cm) can be modified by homothetic transformation while keeping the same mechanical behavior.

We used cohesionless ballistic deposition to generate dense layers (Kadau and Herrmann, 2011) as typically found in snow slab layers. For these layers we used a particle radius of $r = 11 \pm 1$ mm (uniform distribution). The radius variation was introduced to prevent close packing, resulting in a porosity of 45%. Layer density (ρ) was adjusted by changing the particle

density. The size of the particles is not intended to represent the real snow grains. The particle size was chosen as a trade-off between an acceptable computation time (min to day) and avoiding particle size effects in the numerical experiments. At the defined particles scale (larger than the snow grains) the ice properties (e.g. strength, elastic modulus, Poisson's ratio) cannot be used directly. Therefore, the particle scale can be considered as a mesoscale between the macroscopic scale (sample scale) and the microscale (individual snow grains). Hence, we adjusted the particle density to represent the macroscopic snow

densities in accordance with the macroscopic sample porosity.

To characterize the mechanical behavior of these two types of snow-like layers (weak layer or slab layer), unconfined load-controlled tests were performed. To do so, we added an 'actuator' layer generated by cohesionless ballistic deposition, composed of particles of radius $r = 10$ mm on top of the studied layer (Figure 1, red particles). This layer is defined as a rigid clump (following PFC 3D clump theory) with initially low density. The clump density then increased during the load-

controlled test.



The samples were generated in a box; the box walls were then removed to create unconfined test conditions. To avoid a packing effect at the sidewalls, samples were generated 10 particle radius larger and cutout before the simulation. In order to model macroscopic mechanical behavior of the studied layers, we tuned the particle elastic modulus and the bond strength. A large range of particle elastic modulus and bond strength were tested to characterize the relation between particle parameters and

macroscopic behavior. We assumed that bond strength and particle elastic modulus are independent.

*Load-controlled test*

Load-controlled simulations were performed by increasing the actuator layer density. To avoid a sample size effect (see below), 30 cm × 30 cm samples were generated. Our DEM model does not take into account viscous effects or sintering of snow, therefore the results do not depend on the loading rate (not shown). We chose a high loading rate of 20 kPa s$^{-1}$ simply to reduce

the simulation time but we verified that the loading rate did not affect the results.

*Time step*

The length of the time step was determined as function of the particle properties according to

$$\Delta t \approx r \sqrt{\frac{\rho}{E}} \qquad\qquad\qquad (1)$$

where $\rho$ and $r$ are the smallest particle density and radius, respectively, and $E$ is the largest bond or particle elastic modulus.

Choosing the time step in this manner ensures the stability of the DEM model (Gaume et al., 2015).

*Stress and strain*

The average stress and strain were calculated at the interface between the rigid base layer and the studied layer (Figure 1, violet arrow). Normal stress $\bar{\sigma}_z$ was computed as $\bar{\sigma}_z = F_z/A$ and shear stress as $\bar{\sigma}_x = F_x/A$. Here $F_x$ and $F_z$ are the sum of the contact forces acting on the basal layer in the tangential and normal directions, respectively, and $A$ is the area of the basal layer

over which the stresses were determined. We define the engineering strains as normal strain: $\varepsilon_z = \frac{u_z}{D}$ and shear strain: $\gamma_x = \frac{u_x}{D}$ with the displacement of the actuator $u$ in the z- and x-directions and the thickness $D$ of the studied layer. The macroscopic strength ($\sigma^{th}$) was defined as the maximum stress before catastrophic failure. The macroscopic elastic modulus ($E$) was defined as the derivative of the stress before catastrophic failure on the normal stress-strain curve.

*Fabric tensor*

If the particle arrangement during layer creation is not isotropic, the mechanical quantities of the layer show directional dependency. For any heterogeneous, anisotropic material (e.g. bones, concrete, snow), the fabric tensor characterizes the geometric arrangement of the porous material microstructure. The fabric tensor, referred to here as the contact tensor $C$, is the



volume average of the tensor product of the contact unit normal vectors $\bar{n}$. The $2^{nd}$ order contact tensor coefficients are defined in Ken-Ichi (1984) as:

$$C_{i,j} = \frac{1}{N} \ \sum_{\alpha=1}^{N} n_i^{\alpha} \ n_j^{\alpha} \tag{2}$$

where $N$ is the total number of particle contacts, and $n_i^{\alpha}$ are the normalized projections of the contact with respect to the $x_i$
Cartesian coordinate (Shertzer et al., 2011). The contact tensor $C$ was used to estimate the physical properties of the simulated sample.

## 2.2    Laboratory experiments

For model validation, we used data of cold laboratory experiments obtained with a loading apparatus described in Capelli et al. (2018). They performed load-controlled failure experiments on artificially created, layered snow samples, consisting of a
weak layer of depth hoar crystals between harder layers of finer-grained snow. The load applied on the samples was increased linearly until the sample failed. For more information on the experiments see Capelli et al. (2018). We selected three experiments (Table 1) for validating the numerical simulations. For the validation we focused on the normal strain, since for the experimental shear strain data (measure of the horizontal displacement) the signal-to-noise ratio was too low. Furthermore, due to the method used to load the snow samples, data from the force sensor after failure contained experimental artefacts. To
select the model parameters $E_{particle}$ and $\sigma_{bond}^{th}$, we used the elastic modulus computed as the derivative of the normal stress-strain curve and the strength values from the experiments (Table 1), as well as the relations for strength and modulus derived below. Digital image analysis of the experiments had revealed that the deformation was concentrated in the weak layer (Capelli et al., 2018). We therefore simulated the weak layer with a rigid actuator layer on the top.

## 3    Results

This section first presents the structural properties of the two generated layers. The two generated layers were analyzed based on an unconfined compression test. Then, the link between macroscopic behavior and particle properties is described. Finally, the model setup for the weak layer is validated by comparing numerical mixed-modes loading simulations to experimental data.

### 3.1    Structural properties of generated samples

For the sample used to generate the slab, the coefficients of the contact tensor $C$ were (Eq. 2):

$$C_{slab} = \begin{bmatrix} 0.32 & 0 & 0 \\ 0 & 0.32 & 0 \\ 0 & 0 & 0.35 \end{bmatrix} \tag{3}$$



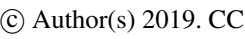

This shows that the slab samples are nearly isotropic, which is in line with results reported for snow types typically found in snow slab layers (Gerling et al., 2017;Srivastava et al., 2016).

For the weak layer samples, 5 in total, the contact tensor was:

$$C_{weak\ layer} = \begin{bmatrix} 0.27 & 0 & 0 \\ 0 & 0.27 & 0 \\ 0 & 0 & 0.46 \end{bmatrix} \tag{4}$$

It shows transverse isotropic symmetry that is again in line with data from snow samples representative for weak layers (i.e. layers of depth hoar, surface hoar or facets), which also show transverse isotropic symmetry (Gerling et al., 2017;Srivastava et al., 2016;Shertzer et al., 2011;Shertzer, 2011).

## 3.2    Characterization of macroscopic properties

*Slab layer*

To establish a relationship between the macroscopic elastic modulus and the particle elastic modulus, we performed 100 simulations (with ten different values of $E_{particle}$ and ten different values of $\sigma_{bond}^{th}$) in pure compression to relate macroscopic and particle parameters. The macroscopic elastic modulus increased linearly with $E_{particle}$:

$$E_{slab\ macro} = \beta_0 + \beta_1\, E_{particle} \tag{5}$$

with the coefficients $\beta_0 = 1.5 \times 10^5$ and $\beta_1 = 0.526$ (Dashed line in Figure 2a; $R^2 = 0.981$).

The macroscopic strength also increased linearly with bond strength

$$\sigma_{slab\ macro}^{th} = \gamma_0 + \gamma_1\, \sigma_{bond}^{th} \tag{6}$$

with the coefficients $\gamma_0 = -318$ and $\gamma_1 = 0.982$) (Dashed line in Figure 2b; $R^2 = 0.999$).

*Weak layer*

For the weak layer we performed 81 simulations (with nine different values of $E_{particle}$ and nine different values of $\sigma_{bond}^{th}$) in
pure compression to relate macroscopic and particle parameters. The macroscopic elastic modulus increased linearly with $E_{particle}$:

$$E_{wl\ macro} = \beta_0 + \beta_1\, E_{particle} \tag{7}$$

with coefficients $\beta_0 = 7.3 \times 10^4$ and $\beta_1 = 0.014$ (Figure 3a; $R^2 = 0.985$).

The macroscopic strength also increased linearly with bond strength





$$\sigma^{th}_{wl\,macro} = \gamma_0 + \gamma_1\,\sigma^{th}_{bond} \tag{8}$$

with coefficients $\gamma_0 = 76.7$ and $\gamma_1 = 0.016$ (Figure 3b; $R^2 = 0.998$).

Hence, based on equations (7) and (8), the macroscopic quantities $\sigma^{th}_{wl\,macro}$ and $E_{wl\,macro}$ can be tuned with $E_{particle}$ and $\sigma^{th}_{bond}$.

### 3.3 Mechanical behavior of layers

*Slab layer*

To investigate the mechanical behavior of the slab layer, we performed load-controlled tension tests. Two phases can be distinguished: linear elastic deformation followed by sample fracture. During the linear elastic deformation, no bond damage appears and the stress linearly increase up to $\varepsilon = 0.0025$ (Figure 4). At failure, the stress dropped rapidly and bond damage drastically increased with increasing strain.

*Weak layer*

The large-deformation, unconfined load-controlled compression tests of weak layer samples revealed four different phases (Figure 5 grey dashed-dotted line). First, there was a linear elastic phase without bond breaking (a.1). When the macroscopic strength was reached, the normal stress dropped sharply during the softening phase as bond damage increased drastically (a.2). During the brittle crushing phase, the sample density as well as the proportion of broken bonds ($P_{broken\,bond}$) steadily increased (a.3). Finally, the densification phase (a.4) was reached at which point the stress increased rapidly while the particles were packed closely together.

By varying the particle modulus $E_{particle}$ and the bond strength $\sigma^{th}_{bond}$ the micromechanical behavior in terms of bond breaking and acceleration (a) of the actuator layer was also investigated more closely up to the start of the brittle crushing phase (Figure 6). Before reaching the macroscopic strength, the normal stress increased linearly with increasing strain while the number of broken bonds and the acceleration were low. The strain at failure depended on both $E_{particle}$ and $\sigma^{th}_{bond}$. During the softening the stress sharply dropped while both the number of broken bonds and the acceleration increased. Both $E_{particle}$ and $\sigma^{th}_{bond}$ controlled the amount of stress drop as well as the rate of increase of $P_{broken\,bond}$ and a. During the brittle crushing phase, both σ and Acc did not change while Bond$_{breaking}$ increased, independent of the values of $E_{particle}$ and $\sigma^{th}_{bond}$.

The stress at the end of the softening phase was characterized by the softening ratio $R = \frac{\hat{\sigma}_{residual}}{\sigma^{th}_{wl\,macro}}$ with $\sigma^{th}_{wl\,macro}$ the macroscopic strength and $\hat{\sigma}_{residual}$ the mean residual stress during the brittle crushing phase. The test with the highest softening ratio (Figure 6 solid light blue line: $R = 0.45$ ) showed the lowest damage and the lowest acceleration. In contrast, the lowest softening ratio (Figure 6 dark blue dashed line: $R = 0.21$) corresponded to the largest proportion of broken bonds





and the largest acceleration. Concerning the two other tests, they exhibited the same residual stress but different softening ratios. We observed that the softening ratio followed a non-linear relation with $E_{particle}$ and $\sigma_{bond}^{th}$.

Similar to the behavior under compression, the mechanical response in shear exhibited different phases: an elastic phase, softening and simultaneously normal brittle crushing and shear displacement and finally shear displacement only (Figure 7 grey dashed-dotted lines). Also the damage dynamics were similar as in pure compression (Figure 7b). No critical bond breaking was observed during the linear elastic phase followed by catastrophic damage after failure. Subsequently, the damage further increased during the brittle crushing. The normal strain increased during the brittle crushing phase and did not change thereafter. The normal deformation was closely related to the proportion of broken bonds, similar to behavior in the pure compression. Shear and normal accelerations reached their maximum at the end of the softening phase (Figure 7c) as observed in pure compression (Figure 6). During the brittle crushing phase, the normal acceleration decreased due to the creation of new contacts that decelerate the actuator layer. The shear acceleration did not change much during the shear displacement phase.

*Weak layer failure envelope*

Unconfined load-controlled tests with nine loading angles were performed to create the failure envelope. Figure 8 complies the values of macroscopic strength for different loading angles resulting in a failure envelope including tension (negative normal stress), pure shear, pure compression as well as mixed-mode loading states. To investigate the influence of sample size, we performed a sensitivity analysis by varying the sample size from 0.1 m × 0.1 m to 1 m × 0.6 m and the random deposition (different ball position generation for the ballistic deposition). Apart from the smallest sample, all samples had very similar failure envelopes, which were fitted with 3$^{rd}$ order polynomial with coefficients $\beta_0^{FE} = -7.2 \times 10^2$, $\beta_1^{FE} = -0.27$, $\beta_2^{FE} = 1.82 \times 10^{-4}$ and $\beta_3^{FE} = 9.53 \times 10^{-9}$ (dash-doted black line in Figure 8, $R^2 = 0.988$):

$$\tau^{th} = \beta_3^{FE} \sigma^{th3} + \beta_2^{FE} \sigma^{th2} + \beta_1^{FE} \sigma^{th} + \beta_0^{FE} \qquad (9)$$

For a sample length of 0.3 m or larger, no effect of sample size on the failure envelope was observed. The sample heterogeneity induced by different types of random deposition did not influence the failure envelope either. Given the expression for the macroscopic strength (Eq. 9), the failure envelope is directly related to $\sigma_{bond}^{th}$.

As the macroscopic strength $\sigma_{wl}^{th}$ is related to $\sigma_{bond}^{th}$ (Eq. 8), the failure envelope can be scaled by using the scaling factor $\left( \frac{\sigma_{wl}^{th}}{\sigma_{wl\,ref}^{th}} \right)$ :

$$\tau^{th} = (\beta_3^{FE} \sigma^{th3} + \beta_2^{FE} \sigma^{th2} + \beta_1^{FE} \sigma^{th} + \beta_0^{FE}) \frac{\sigma_{wl}^{th}}{\sigma_{wl\,ref}^{th}}, \qquad (10)$$

with $\sigma_{wl\,ref}^{th} = 2650$ Pa, which corresponds to the maximum strength in pure compression (Figure 8). Equation (10) allows deriving the failure envelope for any value of the bond strength $\sigma_{bond}^{th}$ (green dash-dotted lines in Figure 9).



### 3.4 Comparison with experimental data

To validate the behavior of our simulated weak layer samples, we used data from laboratory experiments performed by Capelli et al. (2018) (Table 1). For each of the three experiments with different loading angles the simulated total stress ($\sigma_{tot} = \sqrt{\sigma^2 + \tau^2}$) as function of normal strain ($\varepsilon$) is in good agreement with the experimental results.

### 4 Discussion

We used 3-D discrete element model to study the mechanical behavior of simplified snow samples generated by different ballistic deposition techniques. Cohesive ballistic deposition produced transversally isotropic weak layers with high porosity (80%). Cohesionless ballistic deposition produced isotropic slab layers of lower porosity (45%), in general agreement with key properties of natural snow samples (Shertzer, 2011). The DEM particles do not represent real snow grains, to keep the computational costs reasonable (i.e. ~10 min on a standard personal computer for a sample of 50 cm × 50 cm in size, corresponding approximately to 26500 particles). By varying the DEM particle parameters $E_{particle}$ and $\sigma_{bond}^{th}$ the macroscopic properties can be modified to fit different types of snow.

First, tension tests were simulated to study the behavior of dense slab layers. The results evidenced an almost perfectly brittle behavior in good agreement with the tensile behavior reported by Hagenmuller et al. (2014) and by Sigrist (2006).

The mechanical behavior we observed for our weak layer samples, in particular the four phases (Figure 4) during a load-controlled compression test, were very similar to those reported by Mede et al. (2018) who simulated snow behavior with microstructure-based snow samples. More generally, Gibson and Ashby (1997) also described these four distinct phases for elastic-brittle foam samples .

The unconfined load-controlled tests under mixed-mode loading conditions showed shear behavior in good agreement with previously reported results (Reiweger et al., 2015;Mede et al., 2018;Mulak and Gaume, 2019).

The obtained failure envelopes were qualitatively in good agreement with the Mohr-Coulomb-Cap (MCC) model proposed by Reiweger et al. (2015) and with the ellipsoid (cam clay) model proposed by Gaume et al. (2018). The model qualitatively reproduced the snow failure envelopes found in other numerical studies (Mulak and Gaume, 2019;Mede et al., 2018). In our case, the failure envelope is directly linked to $\sigma_{bond}^{th}$, since any failure envelope can be expressed as a function of $\sigma_{bond}^{th}$. Weak layer failure behavior was not affected by the heterogeneity induced by different types of random ball deposition and by the sample size if the sample size was larger than 0.3 m × 0.3 m. This size is typically found in field tests (PST, ECT; van Herwijnen et al., 2016;Reuter et al., 2015;Bair et al., 2014) and laboratory experiments (Capelli et al., 2018).

Based on these purely numerical investigations, the particle and contact parameters were selected to reproduce the results of cold laboratory experiments with real snow samples (Figure 10). The numerical results were qualitatively in good agreement



for the three loading angles. However, the comparison to the experimental results is hindered by the lack of adequate experimental data. Due to vibrations in the actuator plate, the experimental shear strain data could not be used. Hence, there are no experimental data to validate the post-failure behavior. Still, as shown above, the post-failure behavior was in agreement with results of other numerical studies (e.g., Mede et al., 2018).

We showed that the onset of failure corresponded to a strong increase in the number of broken bonds and in actuator layer acceleration. The maximum acceleration was reached towards the end of the softening phase. In fracture mechanics, the zone where softening occurs is generally referred to as the fracture process zone (FPZ) (Bazant and Planas, 1998). Hence, our findings suggest that slab acceleration may be used to accurately track the crack tip location in the weak layer during crack propagation experiments.

The introduction of the softening ratio ($R$) showed that the stress decrease in softening only depends on particle modulus $E_{particle}$ and bond strength $\sigma_{bond}^{th}$, which allows estimating the maximum acceleration of the actuator layer and the damage dynamics. In the present formulation of our model, the softening ratio is fixed for a given pair of parameters ($E_{particle}$ and $\sigma_{bond}^{th}$).

To limit the number of model parameters we made two assumptions: the contact and the bond elastic modulus are equal and
the bond cohesive and tensile strength are equal. The choice of weak layer creation technique (cohesive ballistic deposition) caused unique structural anisotropy that was reflected in the mechanical behavior and added a limitation on the post-failure behavior and the shape of the failure envelope. Investigating microstructure influence by modifying the porosity or the coordination number as the sticky hard sphere (Gaume et al., 2017a) and/or modifying the assumption on contact/bond elastic modulus would allow us to generate a larger range of stress decrease during the softening phase.

Furthermore, in the future, the influence of the ratio between the bond tensile strength and the bond cohesive strength, and/or the weak layer microstructure on the yield surface might be explored.

The developed simulation tool does not take into account snow sintering processes, as we limited the study to fast loading rates. In the context of a dry-snow slab avalanche formation, this means that we can only study artificially induced cracks due to skiers or explosives. In future, we plan to extend the work larger systems with the objective of studying the micromechanics
of the dynamics of crack propagation. Using the presented tool to model a PST already showed some promising preliminary results (Bobillier et al., 2018).

## 5    Conclusions

Understanding the failure behavior of slab and weak layer independently and characterize the influence of the main parameters is a prerequisite for studying the dynamics of crack propagation leading to the release of a dry-snow slab avalanche.



We developed a mesoscale (between snow grain and slope scale) simulation tool based on 3-D discrete element simulations to generate snow layers of varying properties and investigate micromechanical processes at play during snow failure. Two types of snow layers were generated by ballistic deposition techniques: (1) a uniform snow slab and (2) a porous transversally isotropic weak snow layer. These two types of snow layers are the two main components of dry-snow slab avalanches. The

layers were characterized by a linear relation between particle/contact parameters and macroscopic properties. By deliberately making the choice of not representing the real snow microstructure, the computational time decreases and allows creating relatively large systems.

We found an elastic-brittle mechanical behavior for slab layers in tension. The weak layer behavior under mixed-mode loading included four distinct phases of deformation (elastic, softening, simultaneous brittle crushing and shear displacement and

finally shear displacement) as recently reported in the literature. The weak layer failure envelope, derived from a series of mixed-mode loading simulations under different loading angles, was in good agreement with previous experimental and numerical results. The closed-form failure envelope can be tuned by adjusting the bond strength parameter.

Weak layer features analysis showed some of the limitations induced by our assumptions on particle parameters and the uniqueness of the microstructure generation. Still, the validation results suggest that the presented simulation tool can

reproduce the main behavior of weak layers under mixed-mode loading conditions – even though we strongly simplified the microstructure to limit the computational costs.

In the future, we intend to increase the system size and simulate a propagation saw test and explore the dynamics of crack propagation that eventually leads to dry-snow slab avalanche release.

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



**Table 1: Characteristics of the three cold laboratory experiments used for model validation.**

| Characteristics | Experiment | | |
|---|---|---|---|
| | 1 | 2 | 3 |
| Base layer density (kg m$^{-3}$) | 392 | 271 | 289.5 |
| Weak layer density (kg m$^{-3}$) | 174 | 170 | 170 |
| Slab layer density (kg m$^{-3}$) | 399 | 212 | 293 |
| Base layer main type of grain | Faceted crystals | Faceted crystals | Faceted crystals |
| Weak layer main type of grain | Depth hoar | Depth hoar | Depth hoar |
| Slab layer main type of grain | Faceted crystals | Faceted crystals | Faceted crystals |
| Base layer, size of grain (mm) | 0.7-1.5-2 | 1-2 | 1-1.5 |
| Weak layer, size of grain (mm) | 2-4 | 2-4 | 3-4 |
| Slab layer, size of grain (mm) | 0.7-1.5 | 1-2 | 1-1.5 |
| Failure stress (kPa) | 10.5 | 3.2 | 8.3 |
| Failure strain | 0.0019 | 0.00243 | 0.00198 |
| Loading rate stress (Pa s$^{-1}$) | 168 | 168 | 168 |
| Loading angle (°) | 0 | 15 | 35 |



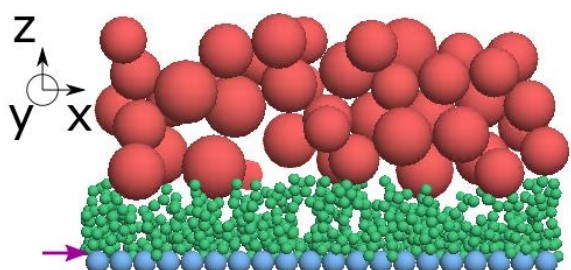

**Figure 1: System coordinates and slice of the generated system consisting of the basal layer (blue), the layer studied, in this case a weak layer, (green) and the actuator layer (red). The violet arrow points to the interface between basal and study layer where the stress is measured.**

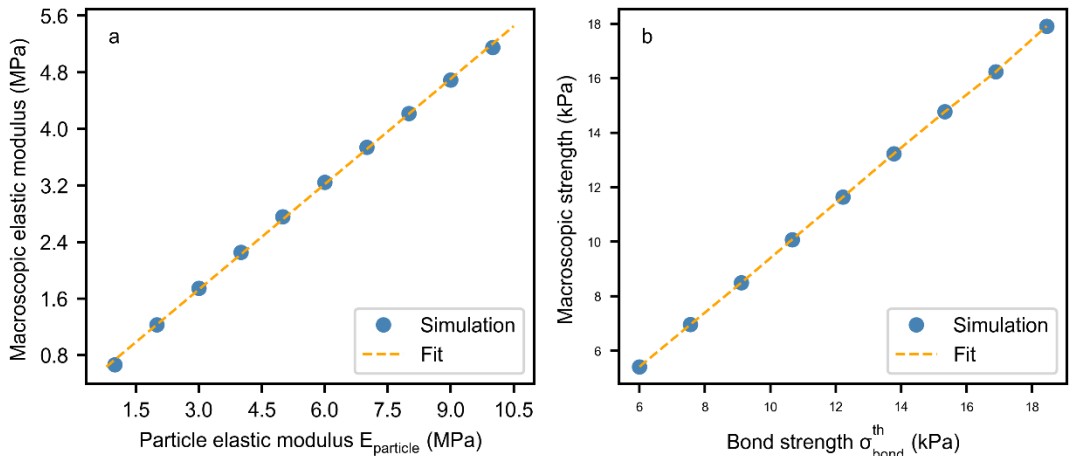

**Figure 2: (a) Slab macroscopic elastic modulus as a function of particles elastic modulus. The blue dots correspond to the mean of ten simulations with different values of $\sigma_{bond}^{th}$. (b) Slab macroscopic strength as a function of slab particles strength obtained with unconfined load-controlled compression simulations. The blue dots correspond to the mean of ten simulations with different values of $E_{particle}$.**



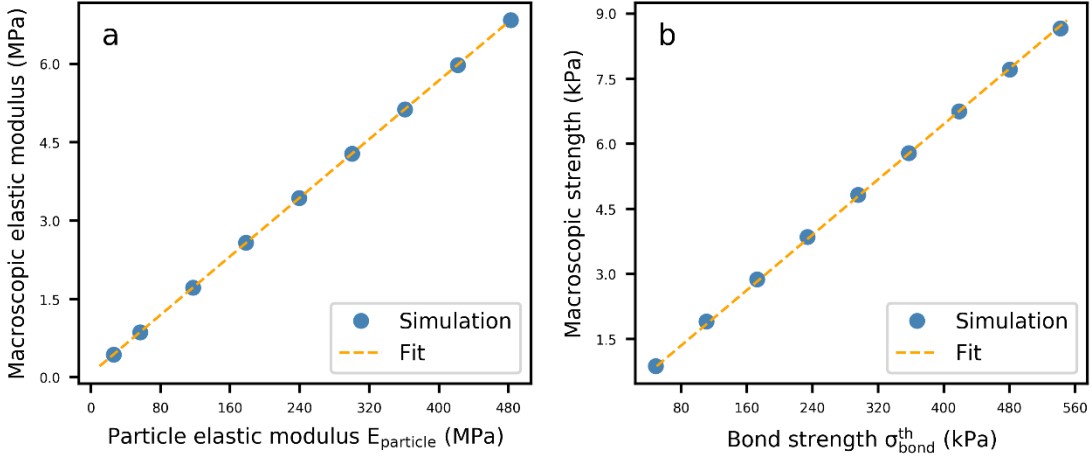

**Figure 3: (a)** Weak layer macroscopic elastic modulus as a function of particles elastic modulus. The blue dots correspond to the mean of nine simulations with different values of $\sigma_{bond}^{th}$. **(b)** Weak layer macroscopic strength as a function of slab particles strength obtained with unconfined load-controlled compression simulations. The blue dots correspond to the mean of nine simulations with different values of $E_{particle}$.

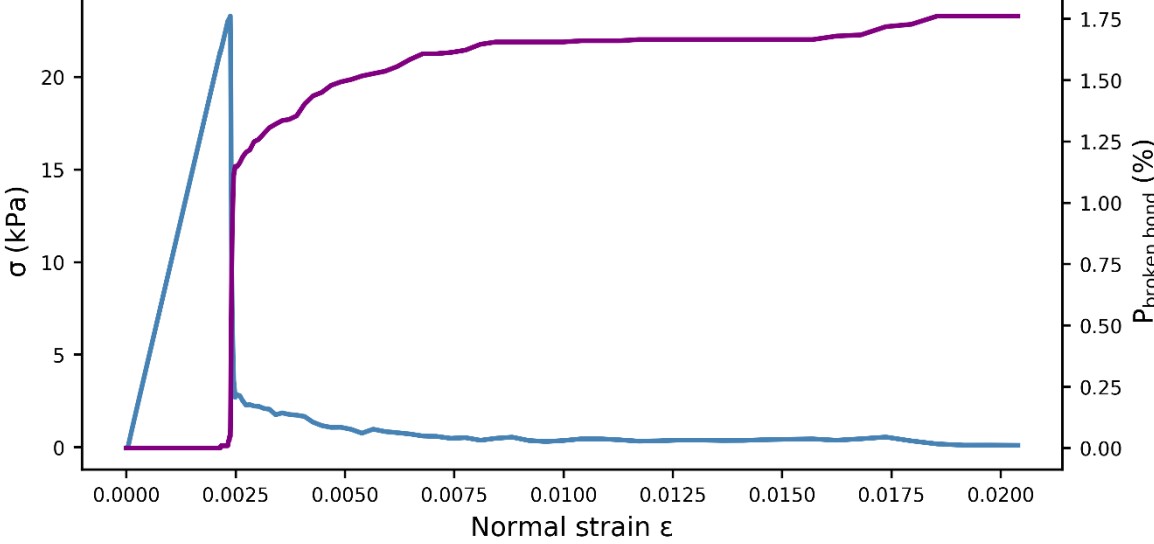

**Figure 4:** Slab layer behavior under load-controlled tension test. The blue line shows the normal stress, the violet line corresponds to the bond breaking ratio are shown as functions of the normal strain.





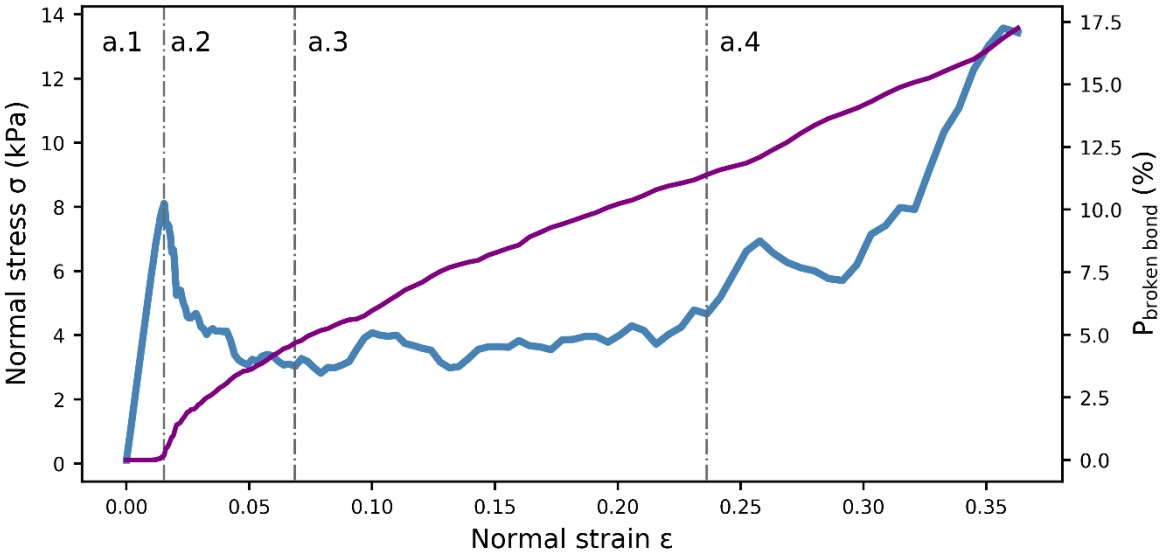

**Figure 5: Weak layer behavior under load-controlled compression test ($E_{particle} = 30MPa$ and $\sigma_{bond}^{th} = 500kPa$). The blue line shows the normal stress during the four phases of weak layer failure. It includes the linear elastic phase (a.1), softening (a.2), brittle crushing (a.3), densification (a.4). The violet line corresponds to the proportion of broken bonds.**





**Figure 6: Weak layer behavior under load-controlled compression tests for four combinations of $E_{particle}$ (solid lines) and $\sigma^{th}_{bond}$ (same color, dashed-dotted lines). (a) Normal stress vs. normal strain. (b) Percentage of broken bonds (damage). (c) Acceleration of the actuator layer. The orange dashed-dotted line represents the approximate beginning of the brittle crushing phase. The grey dotted line represents the beginning of the softening phase defined by the strength (grey dot).**




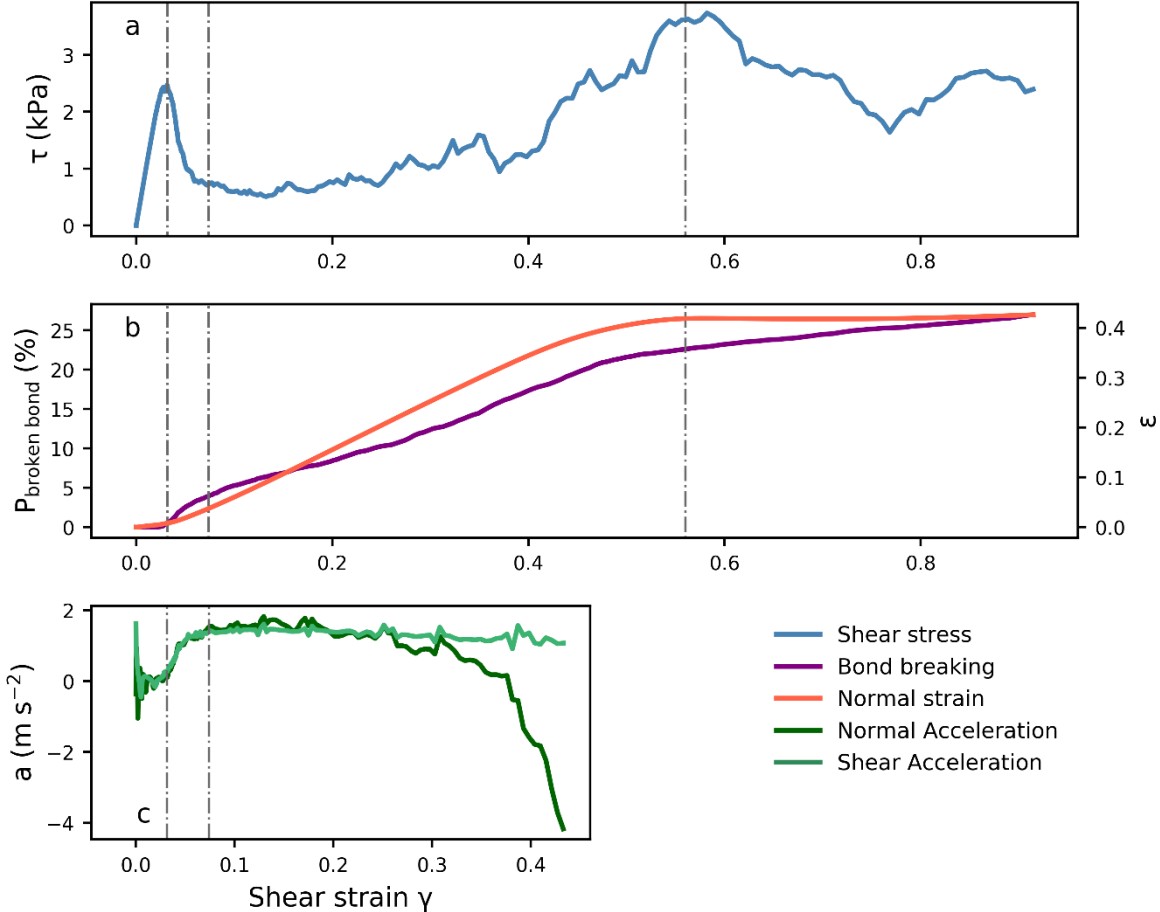

**Figure 7: Weak layer behavior in load-controlled mixed-mode testing at 35° from the horizontal ($E_{particle} = 30MPa$ and $\sigma_{bond}^{th} = 500kPa$). (a) Shear stress, (b) bond damage (violet) and normal strain (orange, right scale), and (c) normal and shear accelerations are shown as function of the shear strain.**





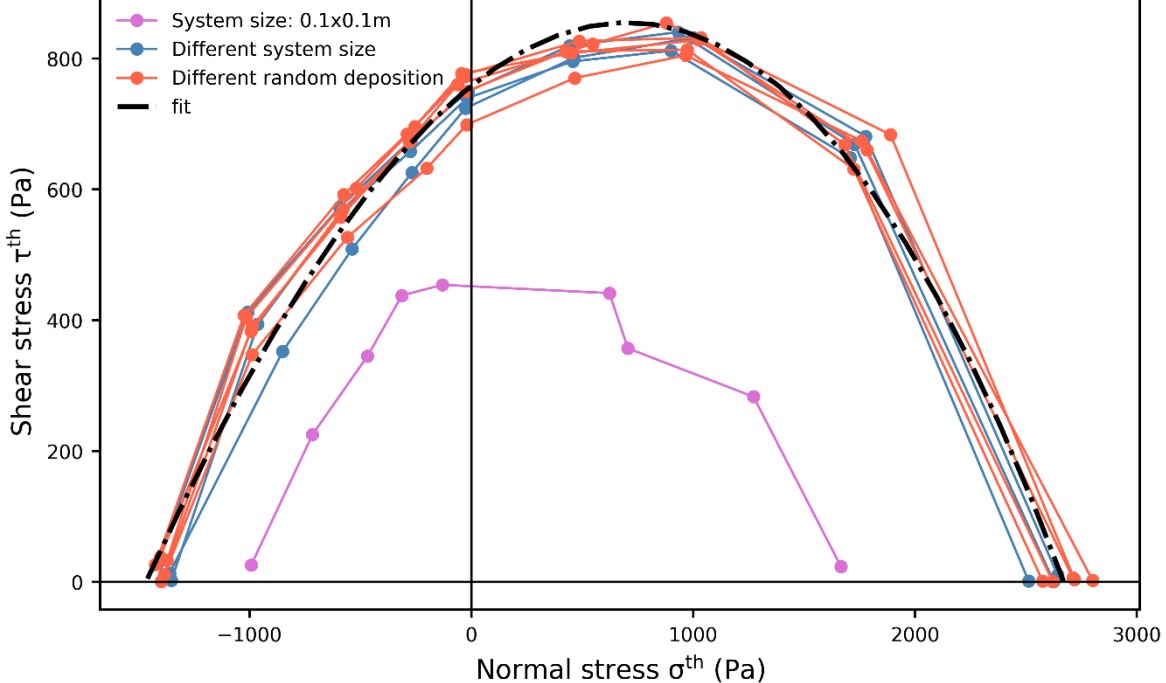

**Figure 8: Failure envelopes for different sample sizes, and types of random particle deposition. The blue lines correspond to different sample sizes from 0.3 m × 0.3 m to 0.6 m × 1 m. The pink line corresponds to a sample size of 0.1 m × 0.1 m. The orange lines correspond to a sample size of 0.3 m × 0.3 m generated with different random depositions. The black dash-dotted line corresponds**
5 **to a 3rd order polynomial fit of all data apart from those obtained with the sample size of 0.1 m × 0.1 m.**





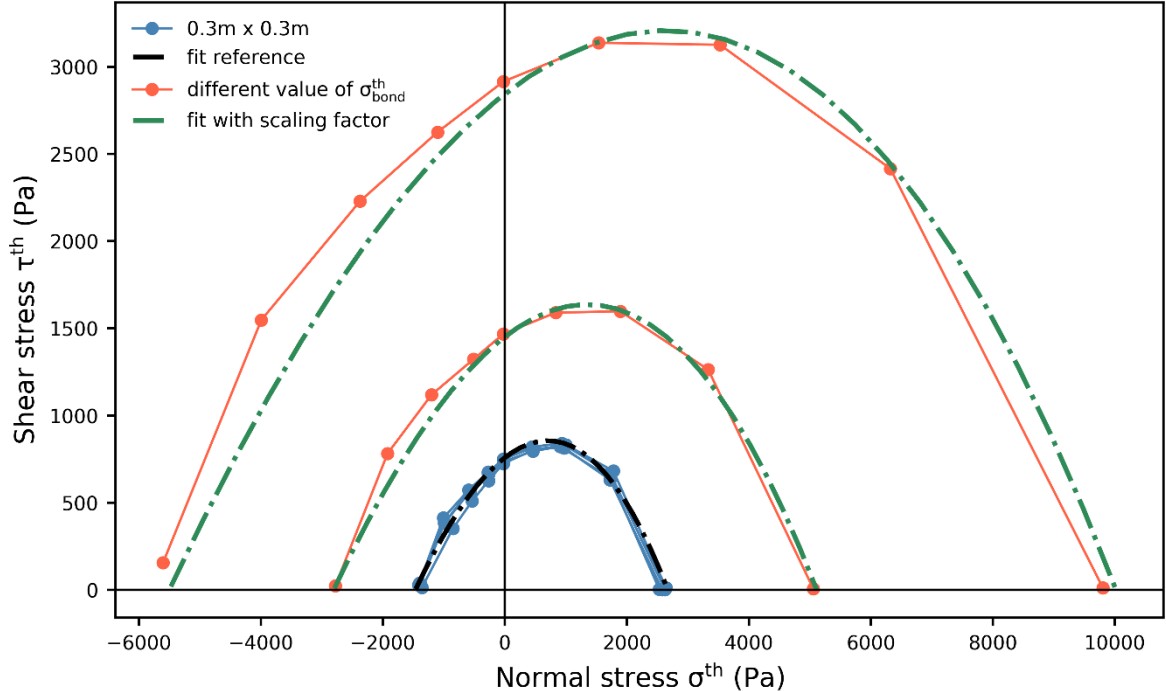

**Figure 9: Failure envelopes for different values of bond strength $\sigma_{bond}^{th}$ and fit only based on equation (10). The blue lines correspond to the data shown in Figure 8 and the black dash-dotted line to the corresponding fit (Eq. 8). The orange lines correspond to failure envelopes with different values of bond strength $\sigma_{bond}^{th}$. The green dash-dotted line corresponds to the corresponding fit defined in equation (10) which do not depend on orange line data.**



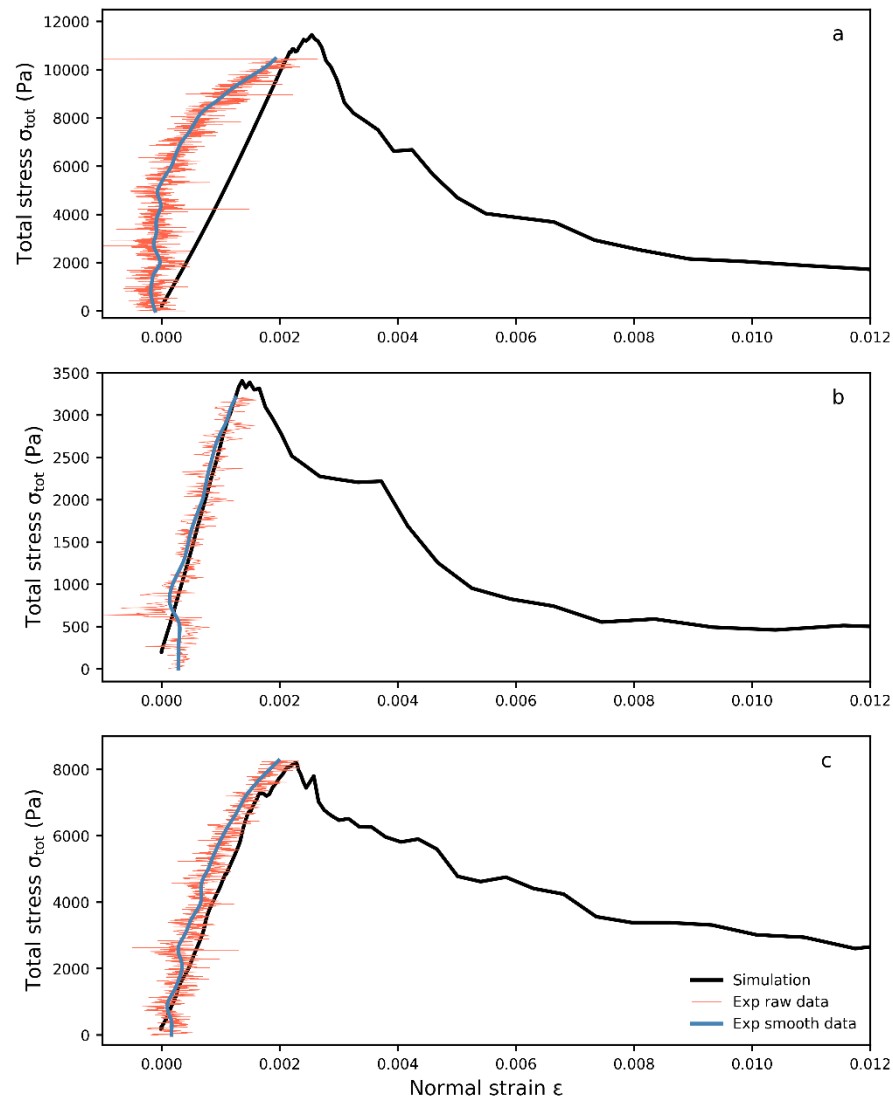

**Figure 10: Total stress as function of normal strain for three simulations and the corresponding experimental results. (a) for a loading angle of 0°, (b) 15° and (c) 35°. The orange lines shows the raw stress data, the blue lines are the smoothed stress using a Kalman filter (Capelli et al., 2018) and the black lines are the simulation results.**

