# Peer review of "Micromechanical modeling of snow failure"

_The Cryosphere, 2019_

## Referee Comment (RC1) · Anonymous Referee #1 · 20 Jun 2019

The manuscript titled "Micromechanical modeling of snow failure" by Bobillier et al. reports the setup and the results of numerical DEM models aiming at studying the failure of weak snow layers. The paper is based on the fact that complex and detailed numerical models are extremely time-consuming. On the contrary, it is possible to build simplified models that are able to catch the main characteristics of the investigated material. In the proposed approach, such models are constituted by different layers of spherical particles. The approach is original and interesting results can be obtained from such numerical setup. The authors "tuned" particle properties by simulating real experiments. This approach is commonly used in other engineering disciplines.

In addition, they predicted the behaviour of such complex material under particular stress conditions, say, pure traction, for which no experimental pieces of evidence are

present. Referring to this last point, the possibility of "extrapolating" the behaviour to something that is hard to replicate in a laboratory has to be further discussed in detail and the limitation of the approach must be clearly stated.

In addition, there are some points that are not clear and must be detailed.

- P.2, L.6: to which properties do the authors refer with "and possibly other ones"?

- Referring to the contact model (P.3), it is not clear when the contacts are activated and when not. In other words, it is possible that new contacts form during the test, or not?

- P.3 L.15: scaling the size of the layer through homothetic transformation does allow to state that the mechanical properties are conserved? A short but detailed study on scaling laws would be appreciated.

- P.3 L.25: a lot of attention is given to the density. Why? It seems that the results are not density-dependent.

- P.4 L.5: the authors assumed that bond strength and particle elastic modulus are independent. Is this consideration supported by data, observations, previous researches, or is it a hypothesis?

- P.4 L.20: it is not clear the test setup. It seems that the density of the actuator layer is rapidly increased to simulate a normal vertical pressure. Why we should expect shear strains into the weak layer?

- Referring to the characterization of macroscopic properties, the authors performed a Latin hypercube sampling on the values of the elastic modulus of the particle and the strength of the bond and obtained the macro-properties of the slab. Many issues arise: why in Figure 2a only 9 simulation points appear, while

the authors have performed 100 simulations? Are those points related to a particular value of $\sigma_{bond}^{th}$? Are the values of coefficients $\beta_0, \ldots$ feasible/realistic? Please add the units of measure to $\beta_0$ and $\gamma_0$.

- Referring to the mechanical behaviour of layers, it is necessary to define what a failure is. Failure in tension is different from failure in compression or in shear. Referring, for example, to tension tests, how such tests were performed? Have the results of tension tests been compared with tests on real snow? In general, synthetic models are able to "interpolate" rather than "extrapolate".

- P.8 L.11: Which is the meaning of "shear acceleration"?

- Referring to the failure envelope reported in Eqn. (9), what $\sigma^{th}$ does represent? Can the failure envelope be used in a real snowpack on a real slope? In addressing this issue, the authors must consider the fact that their tests were performed in unconstrained lateral conditions, different from boundary conditions that can be observed in a continuous layered snowpack.

- As stated in the introduction, the failure of snow slabs depends on many parameters, such as the fracture energy. Have the authors considered this important parameter in their simulations?

- In granular materials, failure mechanisms presupposes the formation and the subsequent destruction of force chains. Evidence of such behaviour has been observed on real snow tests (De Biagi *et al.*, European J. of Mech. - A/Solids, 74, 26-33, 2019). The observation of such mechanisms in simplified numerical models supports the conclusions. Have the authors noted such behaviours in their tests?

---

## Referee Comment (RC2) · Chris Borstad (Referee) · 26 Jun 2019

This manuscript describes a Discrete Element Model (DEM) study of snow deformation and failure. A commercial DEM software package is used to simulate porous and anisotropic weak layers as well as denser and stronger slab layers. The size and properties of the discrete numerical particles was chosen to represent macroscopic layer properties rather than the size and shape of individual snow grains. Load-controlled numerical simulations were performed on both types of layers, using different loading orientations. The nominal stress-strain response of the simulations is discussed, and a weak layer failure envelope is derived. Comparison is made to the results of three experiments from a cold laboratory study, with generally good agreement between the slope of the stress-strain curves and the failure stress.

[Figure]

I am encouraged by the prospects for using DEM simulations to study aspects of snow mechanics and slab avalanche triggering, and this work is a welcome contribution. Most of my comments relate to issues of clarity and presentation. There is a lot of detail in the manuscript, but I find some aspects that are unclear or unsubstantiated.

Snow is a highly rate-dependent material, although the DEM model does not take into account rate effects such as sintering or viscous deformation. This is acceptable, although I think some further discussion of rate effects is warranted to place the results in context. A target "high" loading rate of 20 kPa/s is chosen for the simulations, and there is mention that verification was made that varying the loading rate did not affect the results (although this is not shown; why/how then was 20 kPa/s chosen?). However, for placing the simulation results in context with experimental results in the literature, it would be worth discussing what types of experimental loading rates are appropriate for comparing with these simulation results.

The simulations are performed with the layer of interest (slab or weak layer) sitting on a rigid base. However, weak layers typically are sandwiched between deformable layers (slab and base are usually stiffer, but still deformable). The stress measurements from the simulations are derived from results at the interface between the layer and the rigid base. How might your results differ if you had a multi-layer scheme with a weak layer sitting on a stiffer, but deformable foundation? It seems to me that this would be more appropriate physically.

P2, L1: describe in a bit more detail what the PST is here, for the benefit of readers that may not be familiar

P2, L18-21: "too high" computational cost is vague here, and I'm skeptical of this statement without further justification. High Performance Computing (HPC) systems allow very large and costly simulations of things like climate, weather, ice sheet dynamics, astrophysics, etc. I'm quite sure that a slope-scale simulation would be feasible on a suitable HPC computing cluster, so you might just need to say that such a simulation

is too costly for a stand-alone personal computer (if indeed this is what you mean), or that the commercial code you're using isn't suited (or licensed) for running on a large cluster.

P3, L1-9: The description of the contact model is a bit vague here. A schematic diagram would be helpful to visualize what the model is simulating at the particle scale and what all these mechanical parameters represent physically. It's okay to direct the reader to previous studies that describe such an approach, to a limit, but there's just not enough information here to adequately understand the contact model.

P3 L14: "highly anisotropic" is vague: what is meant by "highly"? I might suggest removing this and just saying "anisotropic"

P3 L15: "can be modified by homothetic transformation" is vague here. Homothetic transformation should be defined. Is this something that is done in the present study, or just something that "can be" done?

P3 L19-20: I missed what the layer densities were that you simulated, and what particle densities were needed to achieve these layer densities. I suggest an additional table of mechanical properties such as this.

P3 L21: "acceptable computation time" doesn't mean much here without a description of what kind of computer you used (later in the text you mention something about a "standard" personal computer, but this is still too vague).

P3 L29: define "clump theory" and "clump density"

P4 L1: it would be worth justifying why you chose unconfined test conditions rather than confined.

P4 L7: I'm not clear how load-controlled tests were performed by "increasing the actuator layer density." I guess by increasing the density you're increasing the weight/gravitational acceleration that the actuator is applying to the layer? A bit more detail is needed here.

P4 L9-10: Related to comment above, here you specify a loading rate. Is this a target loading rate that you achieve by increasing the actuator density?

P4 L19-20: Is "A" the nominal/total area or some measure of the contact area between the particles that represent the layer and the particles that represent the base?

P4 L22-23: At what point along the stress-strain curve is this tangent modulus calculated? This is a common problem with using a tangent modulus to calculate the elastic modulus, because the stress-strain curve is not usually linear all the way to the peak stress. You mention several times that these curves are linear, but I'm skeptical that this is the case. The stress-strain curves in Figures 4, 5, 6, 7, and 10 seem to show some nonlinearity right before the peak (which is to be expected, and is commonly found in experimental data; I recommend zooming in on these peaks in the figures to show any nonlinearity and bond breakage, even if minimal). Thus it really matters where you calculate a tangent modulus, and it is thus common to use something like a secant modulus at the elastic limit (something like 95% of the peak stress) for determining a more robust elastic modulus.

Equation 2: I think the comma between the "i" and "j" subscripts shouldn't be there in $C\_ij$. A comma typically signifies differentiation in standard summation convention (e.g. $C\_i,j = d/dj\ C\_i$), but this is a tensor product of unit normal vectors.

P5 Laboratory experiments: here you chose three experiments from the Capelli study. The Capellis study looked at rate effects, and used three different loading rates. You have chosen results from their intermediate loading rate. Why? How would your results compare to their experimental results at different loading rates? Contact tensors: I'd like to see what the slab and weak layers look like in detail. It's encouraging to see that the weak layer shows transverse isotropy. I think a figure showing the slab and weak layer assemblies in detail would be a nice addition, perhaps even with some unit normal vectors drawn in to show how you're getting these contact tensor results.

P7 L8-9: I think there is some (slight) nonlinearity right before peak, and the step in

bond breaking ratio confirms this. Even a small amount of nonlinearity is important, as it indicates some damage accumulation prior to failure (and this is again why it's important where you calculate the tangent modulus...).

P7 L24: What is "Acc" and "Bond_breaking"? This seems to be new terminology. I'm also confused as to why you have focused so much on acceleration here. What exactly is the acceleration showing? You previously discuss that your results are not sensitive to loading rate variation, but wouldn't you expect some change in these acceleration curves with different loading rates? Even if the stress-strain curves don't change much?

P8 L5-6: What do you mean by "critical" bond breaking here? The bond breaking curve in Figure 7b is obstructed by the normal strain curve, but I'm again inclined to think that there seems to be some nonlinearity/bond breaking right before failure. I would zoom in on the peaks of the stress/strain curves, perhaps in a subset of these figures.

P8 L13: unclear how you're defining "loading angles" here. Another example of where some schematic diagrams would be helpful.

P8 L19: The polynomial fit represented by Eq. 9 indeed looks good, but a goodness-of-fit measure like $R^2$ is not (in general) applicable for a nonlinear model unless a constant mean function can be embedded in the nonlinear model. It's worth checking how the $R^2$ value is calculated here, since it's not going to be the same definition for a goodness-of-fit as in a linear regression model.

P9 L3: another reference to loading angles here, but the coordinate system for defining the angle hasn't been defined (some additional schematic diagrams will alleviate many of these kinds of comments)

P9 L10: "standard personal computer" should be defined more specifically: what kind of processor, how many cores, what type/amount of memory

P9 L14: The experiments of Sigrist were in bending (to induce tensile failure), and predominantly showed quasi-brittle behaviour with clear nonlinear stress-strain (or loaddisplacement) response prior to failure.

P9 L21-22: I can see a better agreement with the cam clay model, but less so with the Mohr-Coulomb-Cap model proposed by Reiweger, which has a linear portion corresponding to the Mohr-Coulomb criterion which is not present in your results. Perhaps worth discussing in a bit more detail, or justifying why you think there is good agreement here?

---

## Author Comment (AC1) · 16 Sep 2019

**Reply to Referee #1**

We thank the reviewer for the insightful and constructive comments. In the following, we reply point by point on the reviewer's comments. They are shown in italic, our replies are in plain red and manuscript modification that we will make are in blue.

*The manuscript titled "Micromechanical modeling of snow failure" by Bobillier et al. reports the setup and the results of numerical DEM models aiming at studying the failure of weak snow layers. The paper is based on the fact that complex and detailed numerical models are extremely time-consuming. On the contrary, it is possible to build simplified models that are able to catch the main characteristics of the investigated material. In the proposed approach, such models are constituted by different layers of spherical particles. The approach is original and interesting results can be obtained from such numerical setup. The authors "tuned" particle properties by simulating real experiments. This approach is commonly used in other engineering disciplines.*

*In addition, they predicted the behavior of such complex material under particular stress conditions, say, pure traction, for which no experimental pieces of evidence are present. Referring to this last point, the possibility of "extrapolating" the behavior to something that is hard to replicate in a laboratory has to be further discussed in detail and the limitation of the approach must be clearly stated.*

*In addition, there are some points that are not clear and must be detailed.*

We will amend our manuscript to clarify some parts that were unclear to the reviewer and more thoroughly discuss the limitations of the model and how the results can be generalized.

1    *P.2, L.6: to which properties do the authors refer with "and possibly other ones"?*
We will add examples of other snow properties that can be related to the dynamics of crack propagation such as slab porosity, weak layer failure envelope, weak layer elasticity, microstructure:

However, no theoretical framework exists that describes how these mechanical properties and possibly other ones such as weak layer failure envelope, weak layer elasticity or microstructure relate to the dynamics of crack propagation at the slope scale.

2    *Referring to the contact model (P.3), it is not clear when the contacts are activated and when not. In other words, it is possible that new contacts form during the test, or not?*
Thank you for this remark. Once a bond breaks, only particle frictional contact occurs and no new bonds are created (i.e no sintering occurs). This assumption is motivated by the fact that

the strain rate is large and the time scale is seconds during a PST experiment. We will explicitly mention that we do not expect sintering.

In addition, a more detailed description of the contact model will be given in the manuscript with the following figures (a, b).

[Figure]

*Figure a: Representation of the PFC parallel bond model (PBM) used in the simulations. a) Normal mechanical parameter bond and unbonded, where $E_b$ represents the bond elastic modulus, $\sigma_t$ the tensile strength, $E_u$ the contact elastic modulus and $e_u$ the restitution coefficient. b) Shear mechanical parameter bond and unbonded, where $E_b$ represents the bond elastic modulus, $\sigma_s$ the shear strength, $E_u$ the contact elastic modulus, $v_b$ the bond Poisson's ratio and $\mu_u$ the friction coefficient.*

[Figure]

*Figure b: Representation of the bonded behavior of PBM used in the simulations. (a) Bond normal force $N_b$ as a function of the normal interpenetration $\delta_n$ scaled by the bond radius $r_b$. (b) Bond shear force $\|S_b\|$ as a function of tangential interpenetration $\delta_s$ scaled by the bond radius $r_b$. (c) Bond-bending moment $\|M_{b,1}\|$ as a function of bending rotation $\theta_1$ scaled by the bond radius $r_b$. (d) Torsion moment $\|M_{b,2}\|$ as a function of twist rotation $\theta_2$ scaled by the bond radius $r_b$.*

3    *P.3 L.15: scaling the size of the layer through homothetic transformation does allow to state that the mechanical properties are conserved? short but detailed study on scaling laws would be appreciated.*

Performing simulations of weak layers of different thickness generated through homothetic transformation shows the same mechanical results (see figures c, d). The bond strength and elastic modulus are scaled such that the macroscopic mechanical behavior becomes almost exactly the same for different values of weak layer thickness (see figures c, d). This will be clarified in a section that will be added to the supplementary material. The equation to characterize macroscopic properties can be written as:

$$E_{wl\,macro} = (\beta_0 + \beta_1\,E_{particle})/(\frac{h_{wl\,ref}}{h_{wl}})$$

$$\sigma^{th}_{wl\,macro} = (\gamma_0 + \gamma_1\,\sigma^{th}_{bond})/(\frac{h_{wl\,ref}}{h_{wl}})$$

Where $h_{wl\,ref} = 3cm$ and $h_{wl}$ correspond to the new weak layer thickness.

[Figure]

*Figure c: Stress-strain curves for weak layers of different thickness (colors) under load-controlled compression. The blue line shows the reference weak layer with a thickness of 3 cm.*

[Figure]

*Figure d: Failure envelopes for weak layers with different thicknesses (colors) and fits based on equation (9).*

**4    P.3 L.25: a lot of attention is given to the density. Why? It seems that the results are not density-dependent**

In a real snowpack, snow density is a very important parameter, as many mechanical properties scale with density (e.g. Shapiro et al. 1997, van Herwijnen et al., 2016). Snow properties such as $\sigma_{slab\,macro}^{th}$ or $E_{slab\,macro}$ are thus often defined as function of mean slab density. However, slab density directly relates to the load on the weak layer, which influences crack propagation propensity and thus slab avalanche release. The density and the thickness of the slab will determine the load. If the stress state given by this load is outside of the failure envelope of the weak layer, failure can occur. Here, a lot of attention was given to the evaluation of the failure envelope, which defines the critical load. In our simulations, as you correctly mention, weak layer density does not play a crucial role.

**5    P.4 L.5: the authors assumed that bond strength and particle elastic modulus are independent. Is this consideration supported by data, observations, previous researches, or is it a hypothesis?**

In some materials strength and elastic modulus are related, while in other materials both properties are not related. For snow, it remains unclear if those two properties are related. Our goal was to independently control both parameters in order to have a precise control on the macroscopic elastic modulus and macroscopic strength of the snow

6    *P.4 L.20: it is not clear the test setup. It seems that the density of the actuator layer is rapidly increased to simulate a normal vertical pressure. Why we should expect shear strains into the weak layer?*

We will improve Figure 1 in the manuscript to more clearly illustrate the test setup (see figure e).

[Figure]

*Figure e: A) Coordinate system and diagram of the setup consisting of the basal layer (blue), the tested layer, in this case a weak layer, (green) and the actuator layer (red). The violet arrow points to the interface between basal and tested layer where the stress is measured. B) slice of a generated system consisting of a slab layer (large red particles) and a porous weak layer (small green particles). A zoom of the weak layer is shown in the circle. The lines represent bonds between particles. Applied gravity is defined on the right where $\psi$ is the loading angle.*

The stress on the weak layer is increased by increasing the density of the actuator layer. By changing the gravity angle, mixed-mode loading is simulated. Through this change in the angle of gravity, the tested layer is under both shear and normal stresses. This will be clarified in the revised manuscript.

7    *Referring to the characterization of macroscopic properties, the authors performed a Latin hypercube sampling on the values of the elastic modulus of the particle and the strength of the bond and obtained the macro-properties of the slab. Many issues arise: why in Figure 2a only 9 simulation points appear, while the authors have performed 100 simulations? Are those points related to a particular value of $\sigma_{bond}^{th}$? Are the values of coefficients $\beta_0$: feasible/realistic? Please add the units of measure to $\beta_0$ and $\gamma_0$.*

Thanks for this remark. We performed 9 x 9 simulations (81 simulations) for the weak layer (Fig. 3) and 10 x 10, 100 simulations for the slab layer (Fig. 2). This will be clarified. In Figure 2a, each blue dot represents the mean value of the macroscopic elastic modulus for a fixed $E_{\text{particle}}$ and ten $\sigma_{bond}^{th}$ values (as explained in the caption). A qualitative comparison with data presented by Shapiro et al. (1997) suggests that the values we used are realistic for slab and weak layers. The goal of the layer characterization is to show that the macroscopic properties ($E_{macro}$, $\sigma_{macro}^{th}$) can be controlled in a range that is typical for snow. The units of $\beta_0$ and $\gamma_0$ are Pascal (Pa), which we will add in the manuscript.

8    *Referring to the mechanical behavior of layers, it is necessary to define what a failure is. Failure in tension is different from failure in compression or in shear. Referring, for example, to tension tests, how such tests were performed? Have the results of tension tests been compared with tests on real snow? In general, synthetic models are able to "interpolate" rather than "extrapolate".*

In our simulations, mixed-mode loading is applied by changing the angle of gravity. We agree that this was not clearly stated. We will revise the manuscript to clarify this point. Hence, the identification or definition of failure does not depend on the mode of loading. Failure is simply identified as the point of maximum shear or normal stress. This point precedes the onset of softening (see Figure below). We will clarify this as well in the revised manuscript.

Tension tests were simulated with a negative gravity ($\psi = 180°$). For tension test results we are in the upper range of the tensile strength values reported by Sigrist (2006). However, we can calibrate our microscopic properties to get the macroscopic properties we want.

9    *P.8 L.11: Which is the meaning of "shear acceleration"?*

Thanks for this remark. We agree that the term shear acceleration was not very adequate and we will change it to "tangential acceleration" in the revised manuscript. This tangential acceleration is the 2$^{nd}$ derivative of the tangential displacement.

10   *Referring to the failure envelope reported in Eqn. (9), what $\sigma^{th}$ does represent? Can the failure envelope be used in a real snowpack on a real slope? In addressing this issue, the authors must consider the fact that their tests were performed in unconstrained lateral conditions, different from boundary conditions that can be observed in a continuous layered snowpack.*

Thank you very much for noticing this omission of defining $\sigma^{th}$ and $\tau^{th}$. $\sigma^{th}$ represents the fitted normal strength and $\tau^{th}$ the fitted shear strength. This will be modified in the revised manuscript. Concerning the second aspect on the confinement and real slopes, we checked that unconfined and confined loading conditions yield the same results for weak layer behavior. This finding is due to the large porosity (80%) of the weak layer (see figure f). We will add this figure to the supplementary material.

[Figure]

*Figure f: Weak layer behavior under load-controlled compression ($E_{particle} = 1$ MPa and $\sigma_{bond}^{th} = 5$ kPa). The blue line shows the normal stress during confined test and the orange line during unconfined test conditions.*

11  *As stated in the introduction, the failure of snow slabs depends on many parameters, such as the fracture energy. Have the authors considered this important parameter in their simulations?*

In our simulations, the fracture energy is related to the area below the stress/strain curve in the softening region. This is a result and not an input of the simulation. This fracture energy depends on the weak layer microstructure, the elastic modulus and the strength. We analyzed the results with regard to the softening ratio instead of the fracture energy which shows that the maximum acceleration and the percentage of broken bond at the end of the softening phase is driven by the softening ratio. To answer your question more specifically, the fracture energy is included in our model but we preferred to analyze our results in terms of strength of materials rather than in terms of fracture mechanics. Gaume et al. (2014) showed that the two approaches can be related to each other.

12 *In granular materials, failure mechanisms presuppose the formation and the subsequent destruction of force chains. Evidence of such behavior has been observed on real snow tests (De Biagi et al., European J. of Mech. - A/Solids, 74, 26-33, 2019). The observation of such mechanisms in simplified numerical models support the conclusions. Have the authors noted such behaviors in their tests?*

For computational reasons, the weak layer was modeled with around 10 vertical particles, which did not allow us to observe a clear strain localization within the weak layer. However, given for the large amount of softening we observe after failure, we expect that higher weak layer resolution would allow to observe this feature.

The figure below shows the stress–strain curve shortly before and after the peak stress; some non-linear behavior appears before failure (figure g: a.1). Following the bond-breaking position before failure (figure h) confirms the presence of initial crack formation. Our simplified model does not explicitly show crack growing by clustering; this behavior could be investigated for by increasing the number of particles in the weak layer, which will increase the vertical resolution.

[Figure]

*Figure g: Zoom of the weak layer behavior under load-controlled compression around sample failure ($E_{particle} = 30MPa$ and $\sigma_{bond}^{th} = 500kPa$). The blue line shows the normal stress before (a.1) and after (a.2) failure of the weak layer. The violet line corresponds to the proportion of broken bonds (%).*

[Figure]

*Figure h: Pre-failure crack formation. Plots (top to bottom) represent stress–strain curves, top views of the position of broken/breaking bonds in the weak layer and a side view of the position of broken/breaking bonds. Red dots represent breaking bonds and blue dots broken bonds. a. weak layer behavior at the time where the stress-strain non-linearity start (shortly before failure). b. weak layer behavior during stress-strain non-linearity. c. weak layer behavior immediately before the failure. Bonds break in spatially random manner (no localization observed)*

**References**

Gaume, J., Schweizer, J., van Herwijnen, A., Chambon, G., Reuter, B., Eckert, N., and Naaim, M.: Evaluation of slope stability with respect to snowpack spatial variability, J. Geophys. Res., 119, 1783-1799, https://doi.org/10.1002/2014JF00319, 2014.

Shapiro, L. H., Johnson, J. B., Sturm, M., and Blaisdell, G. L.: Snow mechanics - Review of the state of knowledge and applications, US Army Cold Regions Research and Engineering Laboratory, Hanover, N.H., U.S.A.CRREL Report 97-3, 43, 1997.

Sigrist, C.: Measurement of fracture mechanical properties of snow and application to dry snow slab avalanche release, Department of Mechanical and Process Engineering, ETH Zurich, Zurich, Switzerland, 139 pp., 2006.

---

## Author Comment (AC2) · 16 Sep 2019

**Reply to Referee #2**

We thank the reviewer for the insightful and constructive comments. In the following, we reply point by point on the reviewer's comments. They are shown in italic, our replies are in plain red and manuscript modification that we will make are in blue.

*This manuscript describes a Discrete Element Model (DEM) study of snow deformation and failure. A commercial DEM software package is used to simulate porous and anisotropic weak layers as well as denser and stronger slab layers. The size and properties of the discrete numerical particles was chosen to represent macroscopic layer properties rather than the size and shape of individual snow grains. Load-controlled numerical simulations were performed on both types of layers, using different loading orientations. The nominal stress-strain response of the simulations is discussed, and a weak layer failure envelope is derived. Comparison is made to the results of three experiments from a cold laboratory study, with generally good agreement between the slope of the stress-strain curves and the failure stress.*

*I am encouraged by the prospects for using DEM simulations to study aspects of snow mechanics and slab avalanche triggering, and this work is a welcome contribution. Most of my comments relate to issues of clarity and presentation. There is a lot of detail in the manuscript, but I find some aspects that are unclear or unsubstantiated.*

Thank you very much for this positive appreciation of our work. Concerning the clarity of the paper, we will substantially modify our manuscript to take that comment into account. Please refer to the detailed replies below.

*Snow is a highly rate-dependent material, although the DEM model does not take into account rate effects such as sintering or viscous deformation. This is acceptable, although I think some further discussion of rate effects is warranted to place the results in context. A target "high" loading rate of 20 kPa/s is chosen for the simulations, and there is mention that verification was made that varying the loading rate did not affect the results (although this is not shown; why/how then was 20 kPa/s chosen?). However, for placing the simulation results in context with experimental results in the literature, it would be worth discussing what types of experimental loading rates are appropriate for comparing with these simulation results.*

*The simulations are performed with the layer of interest (slab or weak layer) sitting on a rigid base. However, weak layers typically are sandwiched between deformable layers (slab and base are usually stiffer, but still deformable). The stress measurements from the simulations are derived from results at the interface between the layer and the rigid base. How might your results differ if you had a multi-layer scheme with a weak layer sitting on a stiffer, but deformable foundation? It seems to me that this would be more appropriate physically.*

Concerning the general comment above, we will discuss the hypothesis of analyzing a single layer based on the results of the concentration of deformation in the weak layer as e.g. was shown by Capelli et al. (2018) in more detail in the revised manuscript. Below, we reply to the specific points raised.

1   *P2, L1: describe in a bit more detail what the PST is here, for the benefit of readers that may not be familiar.*

As suggested, we will add a short description in the revised manuscript:

Our understanding of crack propagation was greatly improved by the introduction of the Propagation Saw Test (PST; Gauthier and Jamieson, 2006; Sigrist and Schweizer, 2007; van Herwijnen and Jamieson, 2005). The PST involves isolating a snow column and initiating a crack by cutting in a pre-defined weak layer until the critical crack length is reached and self-propagation starts. The PST allows analyzing the onset and dynamics of crack propagation and deriving important mechanical properties using particle tracking velocimetry (e.g., van Herwijnen et al., 2016).

2   *P2, L18-21: "too high" computational cost is vague here, and I'm skeptical of this statement without further justification. High Performance Computing (HPC) systems allow very large and costly simulations of things like climate, weather, ice sheet dynamics, astrophysics, etc. I'm quite sure that a slope-scale simulation would be feasible on a suitable HPC computing cluster, so you might just need to say that such a simulation is too costly for a stand-alone personal computer (if indeed this is what you mean), or that the commercial code you're using isn't suited (or licensed) for running on a large cluster.*

Simulating a load-controlled compression test for a weak layer sample of 30 cm x 30 cm requires 19,000 DEM particles. The computational costs of such a simulation on a standard personal computer (intel i7 8 processors 3.4Ghz, RAM 16Go) is 1-2 h depending on the mechanical parameters. Using a more powerful system (intel xeon 28 processors 2.6Ghz, RAM: 256Go) reduces this time to 20-30 min. The total simulation time includes two parts: about 70% of the simulation time is used to solve the DEM equation where the time step is determined as function of the particles-contacts properties according to: $\Delta t \approx r \sqrt{\frac{\rho}{E}}$, where $E = 40 - 480\ MPa$; this means that the DEM time step is between $10^{-6}\ to\ 10^{-8}$ s. The remaining about 30% of the simulation time is required to loop on contacts or particles to extract mechanical results: this looping is not parallelized on PFC software.

3    P3, L1-9: The description of the contact model is a bit vague here. A schematic diagram would be helpful to visualize what the model is simulating at the particle scale and what all these mechanical parameters represent physically. It's okay to direct the reader to previous studies that describe such an approach, to a limit, but there's just not enough information here to adequately understand the contact model.

We agree and will describe the contact model in more detail and add the following two figures in the revised manuscript.

[Figure]

*Figure a: Representation of the PFC parallel bond model (PBM) used in the simulations. a) Normal mechanical parameter bond and unbonded, where $E_b$ represents the bond elastic modulus, $\sigma_t$ the tensile strength, $E_u$ the contact elastic modulus and $e_u$ the restitution coefficient. b) Shear mechanical parameter bond and unbonded, where $E_b$ represents the bond elastic modulus, $\sigma_s$ the shear strength, $E_u$ the contact elastic modulus, $v_b$ the bond Poisson's ratio and $\mu_u$ the friction coefficient.*

[Figure]

*Figure b: Representation of the bonded behavior of PBM used in the simulations. (a) Bond normal force $N_b$ as a function of the normal interpenetration $\delta_n$ scaled by the bond radius $r_b$. (b) Bond shear force $\|S_b\|$ as a function of tangential interpenetration $\delta_s$ scaled by the bond radius $r_b$. (c) Bond-bending moment $\|M_{b,1}\|$ as a function of bending rotation $\theta_1$ scaled by the bond radius $r_b$. (d) Torsion moment $\|M_{b,2}\|$ as a function of twist rotation $\theta_2$ scaled by the bond radius $r_b$.*

4    *P3 L14: "highly anisotropic" is vague: what is meant by "highly"? I might suggest removing this and just saying "anisotropic"*

We will remove ''highly'' in the revised manuscript.

5    P3 L15: "can be modified by homothetic transformation" is vague here. Homothetic transformation should be defined. Is this something that is done in the present study, or just something that "can be" done?

By applying a scaling factor on the particle position and particle radius the system can be scaled up or down. The bond strength and elastic modulus are scaled such that the macroscopic mechanical behavior becomes almost exactly the same for different values of weak layer thickness (see figures c,d). A scaling factor needs to be applied to predict the failure envelope and the microscopic-macroscopic relation. The equation to characterize macroscopic properties can be written as:

$$E_{wl\,macro} = (\beta_0 + \beta_1\,E_{particle})/(\frac{h_{wl\,ref}}{h_{wl}})$$

$$\sigma^{th}_{wl\,macro} = (\gamma_0 + \gamma_1\,\sigma^{th}_{bond})/(\frac{h_{wl\,ref}}{h_{wl}})$$

Where $h_{wl\,ref} = 3$ cm and $h_{wl}$ corresponds to the new weak layer thickness.

This will be clarified in a section that will be added to the supplementary material.

[Figure]

*Figure c: Stress-strain curves for weak layers of different thickness (colors) under load-controlled compression. The blue line shows the reference weak layer with a thickness of 3 cm.*

[Figure]

*Figure d: Failure envelopes for weak layers with different thickness (colors) and fit based on equation (9).*

6    *P3 L19-20: I missed what the layer densities were that you simulated, and what particle densities were needed to achieve these layer densities. I suggest an additional table of mechanical properties such as this.*

As suggested, we will add the following table showing the layer properties used in the simulations.

| Mechanical property | Macroscopic | Particles |
|---|---|---|
| Weak layer density (kg m$^{-3}$) | 110 | 550 |
| Slab layer density (kg m$^{-3}$) | 250 | 455 |
| Slab porosity | 45% | - |
| Weak layer porosity | 80% | - |
| Slab elastic modulus | 0.7 – 5.5 MPa | 1 – 10.5 MPa |
| Weak layer elastic modulus | 0.5 – 7 MPa | 40 – 480 MPa |
| Slab strength | 5 – 18 kPa | 6 – 19 kPa |
| Weak layer strength | 1 – 9 kPa | 70 – 560 kPa |

*7    P3 L21: "acceptable computation time" doesn't mean much here without a description of what kind of computer you used (later in the text you mention something about a "standard" personal computer, but this is still too vague).*

Please refer to the reply on comment #2 above.

*8    P3 L29: define "clump theory" and "clump density"*

A clump is a rigid body made of particles and follows rigid body motion (no deformation). The clump theory will be defined in the revised manuscript as follows:

A clump is a rigid collection of $n$ rigid particles that form one DEM element. The volume is defined by the particle positions and radius. The mass properties are defined by the clump density and clump volume. Clumps can translate and rotate but cannot deform. Clump motion obeys the equations of motion induced by the definition of mass properties, loading conditions and velocity conditions.

9    P4 L1: it would be worth justifying why you chose unconfined test conditions rather than confined.

Unconfined conditions were chosen to compare the results to laboratory experiments. We will clarify this in the revised manuscript. We checked that unconfined and confined loading conditions yield the same results for weak layer behavior. This finding is due to the large porosity (80%) of the weak layer (see figure e). We will add this Figure to the supplementary material.

[Figure]

*Figure e: Weak layer behavior under load-controlled compression test ($E_{particle} = 1$ MPa and $\sigma_{bond}^{th} = 5$ kPa). The blue line shows the normal stress during confined test and the orange line during unconfined test conditions.*

10  *P4 L7: I'm not clear how load-controlled tests were performed by "increasing the actuator layer density." I guess by increasing the density you're increasing the weight/gravitational acceleration that the actuator is applying to the layer? A bit more detail is needed here.*

Thank you for your comment; your guess is correct. We will clarify the loading procedure in the manuscript. By increasing the density of the actuator layer (clump density), the stress is increasing.

11  *P4 L9-10: Related to comment above, here you specify a loading rate. Is this a target loading rate that you achieve by increasing the actuator density?*

We compute the simulation loading rate as the normal load ($\rho\ g\ d\ \cos\psi$) divided by the time step. The model is not affected by the loading rate. A loading rate of 20 kPa s$^{-1}$ was chosen to reduce the computational time.

12  *P4 L19-20: Is "A" the nominal/total area or some measure of the contact area between the particles that represent the layer and the particles that represent the base?*

The area "A" is the total area of the base, this will be clarified in the revised manuscript.

13  *P4 L22-23: At what point along the stress-strain curve is this tangent modulus calculated? This is a common problem with using a tangent modulus to calculate the elastic modulus, because the stress-strain curve is not usually linear all the way to the peak stress. You mention several times that these curves are linear, but I'm skeptical that this is the case. The stress-strain curves in Figures 4, 5, 6, 7, and 10 seem to show some nonlinearity right before the peak (which is to be expected, and is commonly found in experimental data; I recommend zooming in on these peaks in the figures to show any nonlinearity and bond breakage, even if minimal). Thus it really matters where you calculate a tangent modulus, and it is thus common to use something like a secant modulus at the elastic limit (something like 95% of the peak stress) for determining a more robust elastic modulus. Equation 2: I think the comma between the "i" and "j" subscripts shouldn't be there in C_ij. A comma typically signifies differentiation in standard summation convention (e.g. C_i,j = d/dj C_i), but this is a tensor product of unit normal vectors.*

To not account for the non-linear behavior shortly before failure, we derive the elastic modulus as the secant modulus up to 70% of the peak stress.

Thanks for your comment, the comma should not be there and will be deleted in the revised manuscript.

14  *P5 Laboratory experiments: here you chose three experiments from the Capelli study. The Capellis study looked at rate effects, and used three different loading rates. You have chosen results from their intermediate loading rate. Why? How would your results compare to their experimental results at different loading rates? Contact tensors: I'd like to see what the slab and weak layers look like in detail. It's encouraging to see that the weak layer shows transverse isotropy. I think a figure showing the slab and weak layer assemblies in detail would be a nice addition, perhaps even with some unit normal vectors drawn in to show how you're getting these contact tensor results.*

Indeed, the experiments by Capelli were done for higher loading rates ($400 Pa\ s^{-1}$). The following Figure shows the direct comparison to these experiments. A loading rate of $400\ Pa\ s^{-1}$ causes the shear apparatus to vibrate so that the raw data become noisy. Nevertheless, the qualitative agreement is fairly good.

We choose to analyze the behavior of each layer independently, since Capelli et al. (2018) showed that the deformation was concentrated in the weak layer. We will look into the slab and weak layer assemblies in more detail in our next study on the propagation saw test.

[Figure]

*Figure f: Total stress as function of normal strain for three simulations and the corresponding experimental results  done by Capelli following the procedure describe by Capelli et al. (2018) at loading rate = 400Pa/s*

15 *P7 L8-9: I think there is some (slight) nonlinearity right before peak, and the step in bond breaking ratio confirms this. Even a small amount of nonlinearity is important, as it indicates some damage accumulation prior to failure (and this is again why it's important where you calculate the tangent modulus...).*

As you assume, there is some non-linear behavior shortly before the stress peak. The figure below shows the stress–strain curve shortly before and after peak stress (failure); we will add and discuss this figure in the revised manuscript.

[Figure]

*Figure g : Zoom of the weak layer behavior under load-controlled compression around sample failure ($E_{particle} = 30MPa$ and $\sigma_{bond}^{th} = 500kPa$). The blue line shows the normal stress before (a.1) and after (a.2) failure of the weak layer. The violet line corresponds to the proportion of broken bonds (%).*

16 P7 L24: What is "Acc" and "Bond_breaking"? This seems to be new terminology. I'm also confused as to why you have focused so much on acceleration here. What exactly is the acceleration showing? You previously discuss that your results are not sensitive to loading rate variation, but wouldn't you expect some change in these acceleration curves with different loading rates? Even if the stress-strain curves don't change much?

Thanks for noting this mistake. "Acc" corresponds to acceleration and "Bond_breaking" to the percentage of broken bonds ($P_{broken\ bonds}$).

We choose to focus on the acceleration because the acceleration steeply increases at failure. We assume that this information will be useful to define the crack tip location during the crack propagation process. The differences with regard to the acceleration plateau are related to the softening ratio, which may be an important indicator for the dynamics of crack propagation.

17   *P8 L5-6: What do you mean by "critical" bond breaking here? The bond breaking curve in Figure 7b is obstructed by the normal strain curve, but I'm again inclined to think that there seems to be some nonlinearity/bond breaking right before failure. I would zoom in on the peaks of the stress/strain curves, perhaps in a subset of these figures.*

The critical bond-breaking defines the large increase of breaking bonds corresponding to catastrophic damage. A figure showing the non-linearity will be added in the revised manuscript.

[Figure]

*Figure h: a. Slab layer behavior under load-controlled tension. The blue line shows the normal stress, the violet line corresponds to the bond breaking ratio are shown as functions of the normal strain. b. Zoom around the stress peak.*

18   *P8 L13: unclear how you're defining "loading angles" here. Another example of where some schematic diagrams would be helpful.*

Thanks for your comment. We will improve Figure 1 in the revised manuscript to more clearly illustrate the test setup (see below).

[Figure]

*Figure i: A) Coordinate system and diagram of the setup consisting of the basal layer (blue), the tested layer, in this case a weak layer, (green) and the actuator layer (red). The violet arrow points to the interface between basal and tested layer where the stress is measured. B) slice of a generated system consisting of a slab layer (large red particles) and a porous weak layer (small green particles. A zoom of the weak layer is shown in the circle. The lines represent bonds between particles. Applied gravity is defined on the right where $\psi$ is the loading angle.*

19  P8 L19: The polynomial fit represented by Eq. 9 indeed looks good, but a goodness-of- fit measure like $R^2$ is not (in general) applicable for a nonlinear model unless a constant mean function can be embedded in the nonlinear model. It's worth checking how the $R^2$ value is calculated here, since it's not going to be the same definition for a goodness-of-fit as in a linear regression model.

Thanks for your remark, the goodness-of-fit measure of this polynomial fit will be modified in the revised manuscript.

20  P9 L3: another reference to loading angles here, but the coordinate system for defining the angle hasn't been defined (some additional schematic diagrams will alleviate many of these kinds of comments) P9 L10: "standard personal computer" should be defined more specifically: what kind of processor, how many cores, what type/amount of memory

These definitions and clarifications will be added in the revised manuscript. Please see the replies to the comments #2 and #17 above.

21  *P9 L14: The experiments of Sigrist were in bending (to induce tensile failure), and predominantly showed quasi-brittle behaviour with clear nonlinear stress-strain (or load-displacement) response prior to failure.*

As shown above in the replies to comments #15 (weak layer) and #16 (slab), there is indeed some non-linear behavior before failure. We will discuss the non-linear behavior shortly prior to failure in more detail in the revised manuscript.

22  P9 L21-22: I can see a better agreement with the cam clay model, but less so with the Mohr-Coulomb-Cap model proposed by Reiweger, which has a linear portion corresponding to the Mohr-Coulomb criterion which is not present in your results. Perhaps worth discussing in a bit more detail, or justifying why you think there is good agreement here?

We agree that our yield surface is close to a modified cam clay model similar to the one shown by Gaume et al. (2018). We will modify the text in the revised manuscript as follows:

The obtained failure envelopes were qualitatively in good agreement with the Mohr-Coulomb-Cap (MCC) model proposed by Reiweger et al. (2015) and with the ellipsoid (cam clay) model proposed by Gaume et al. (2018) and Mede at al. (2018).

**References**

Capelli, A., Reiweger, I., and Schweizer, J.: Acoustic emissions signatures prior to snow failure, J. Glaciol., 64, 543-554, https://doi.org/10.1017/jog.2018.43, 2018.

Gaume, J., Gast, T., Teran, J., van Herwijnen, A., and Jiang, C.: Dynamic anticrack propagation in snow, Nat. Commun., 9, 3047, https://doi.org/10.1038/s41467-018-05181-w, 2018.

Gauthier, D., and Jamieson, J. B.: Towards a field test for fracture propagation propensity in weak snowpack layers, J. Glaciol., 52, 164-168, 2006.

Hagenmuller, P., Theile, T. C., and Schneebeli, M.: Numerical simulation of microstructural damage and tensile strength of snow, Geophys. Res. Lett., 41, 86-89, https://doi.org/10.1002/2013gl058078, 2014.

Mede, T., Chambon, G., Hagenmuller, P., and Nicot, F.: Snow failure modes under mixed loading, Geophys. Res. Lett., 45, https://doi.org/10.1029/2018GL080637, 2018.

Reiweger, I., Gaume, J., and Schweizer, J.: A new mixed-mode failure criterion for weak snowpack layers, Geophys. Res. Lett., 42, 1427-1432, https://doi.org/10.1002/2014GL062780, 2015.

Sigrist, C.: Measurement of fracture mechanical properties of snow and application to dry snow slab avalanche release, Department of Mechanical and Process Engineering, ETH Zurich, Zurich, Switzerland, 139 pp., 2006.

Sigrist, C., and Schweizer, J.: Critical energy release rates of weak snowpack layers determined in field experiments, Geophys. Res. Lett., 34, L03502, https://doi.org/10.1029/2006GL028576, 2007.

van Herwijnen, A., and Jamieson, B.: High-speed photography of fractures in weak snowpack layers, Cold Reg. Sci. Technol., 43, 71-82, https://doi.org/10.1016/j.coldregions.2005.05.005, 2005.

van Herwijnen, A., Gaume, J., Bair, E. H., Reuter, B., Birkeland, K. W., and Schweizer, J.: Estimating the effective elastic modulus and specific fracture energy of snowpack layers from field experiments, J. Glaciol., 62, 997-1007, https://doi.org/10.1017/jog.2016.90, 2016.